# Learning Curves in Robotic Urological Oncological Surgery: Has Anything Changed During the Last Five Years?

**DOI:** 10.3390/cancers17081334

**Published:** 2025-04-15

**Authors:** Theodoros Tokas, Charalampos Mavridis, Athanasios Bouchalakis, Chrisoula Maria Nakou, Charalampos Mamoulakis

**Affiliations:** 1Department of Urology, University General Hospital of Heraklion, University of Crete, Medical School, 71500 Heraklion, Greece; ch.mavridis@uoc.gr (C.M.); mpouxalakisth@gmail.com (A.B.); chrisoulanakou@gmail.com (C.M.N.); mamoulak@uoc.gr (C.M.); 2Training and Research in Urological Surgery and Technology (T.R.U.S.T.)-Group, 6060 Hall in Tirol, Austria

**Keywords:** learning curves, robot-assisted surgery, radical prostatectomy, nephrectomy, nephroureterectomy, radical cystectomy

## Abstract

Robotic surgery has conquered the field of surgical oncology in urology. Robotics is as effective as long-standing open surgery, with some advantages regarding pain, length of hospital stay, and recovery. Residents and urologists have less and less exposure to open surgery across the world, with some countries almost wholly leaning toward robotic surgery, especially in the field of radical prostatectomy. The direct and rapid transition to robotics is possible and efficient from laparoscopy and open surgery. Although the learning curves (LCs) for robotic surgery have been studied, they are still vague without a fully structured curriculum. In this work, we reviewed the current literature for available studies regarding the LCs for robotic surgery in surgical oncology in urology. Furthermore, we investigated possible improvements in training by comparing the LCs of the last 5 years to those of previously published results.

## 1. Introduction

Learning curves (LCs), the process by which a surgeon can master a surgical procedure, can vastly differ between individuals [1]. The LC concept in surgery is often discussed; however, there is no consensus on a definition. Researchers have used a variety of parameters—such as operative time (OT), estimated blood loss (EBL), complication rates, or composite measures such as the trifecta—to demarcate the concept of the “LC”, with a resultant heterogeneity of views in the literature. The inconsistencies noted between studies present challenges to comparability between them [2]. Nonetheless, LCs are crucial in surgical outcomes, patient safety, and cost-effectiveness [3]. Robot-assisted radical prostatectomy (RARP) seems to have a very steep LC, and depending on the factor being evaluated, it can reach up to 350 procedures [4]. Robotic partial nephrectomy (RAPN) appears to be very challenging, with a recent study reporting 150 cases as the plateau for warm ischemia time (WIT) but not for postoperative complications [5]. Robotic radical cystectomy (RARC) is trending as an alternative to open surgery [6]. However, the LC is unspecified, and in combination with the complexity of the procedure, it can be steep and have negative consequences for the patients [7]. Our review’s purpose was to systematically search and analyze the literature regarding LCs in robotic urological oncology procedures (RARP, RAPN, and RARC) to synthesize the documented LC durations and to examine whether LCs over the last five years (2020–2024) do or do not indicate reduction or extension—compared to those of previous periods. Through the analysis of temporal trends, we aimed to assess if technological improvements and the expansion of training programs have impacted the intricacy or length of LCs associated with these surgical techniques.

## 2. Material and Methods

This systematic search was conducted following the Preferred Reporting Items for Systematic Reviews and Meta-analyses (PRISMA) statement [8], using the following search terms: “Robo* AND (cancer OR tumo*) AND (Prostate OR Kidney OR Renal OR Bladder OR Ureter OR Prostatectomy OR Nephrectomy OR Nephroureterectomy OR or Cystectomy) AND (Learning OR Curve OR Learning Curve)”. A comprehensive electronic search was carried out using MEDLINE (Ovid Medline Epub Ahead of Print, In-Process & Other Non-Indexed Citations, Ovid MEDLINE(R) Daily, and Ovid MEDLINE(R)). The literature search was finalized on 1 December 2024. The study search strategy was conducted without any limitation on publication year. We also reviewed the cited references of published systematic reviews and the included studies.

After excluding duplicate records, citations in abstract form, and non-English citations from the final literature search for eligibility, the titles and abstracts of full papers were screened for relevance, defined as original research focusing on the topic “Learning curves in robotic surgery for urologic cancer”. Two researchers (A.B. and MC.N.) independently screened the titles and abstracts of identified journals. Next, the same researchers independently screened full-text records. Any disagreements were resolved through consensus or consulting a senior team member (T.T). Due to study heterogeneity, we finally summarized the included studies qualitatively and did not perform a quantitative meta-analysis. Hence, this work consists of a systematic search but is a narrative literature review.

Statistical analyses were performed using Python (version 3.10) and the SciPy library (version 1.11), applying independent samples *t*-tests for comparing group means. We conducted a descriptive statistical analysis to compare the average number of cases required to reach a plateau between early studies (published before 2020) and recent studies (published from 2020 onwards). For each outcome across RARP, RAPN, and RARC, mean ± standard deviation (SD) values were calculated separately for early and recent studies. An independent samples *t*-test was then applied to assess statistical significance between the two periods, with a *p*-value < 0.05 considered significant.

## 3. Results

Eighty-two studies evaluated the LCs of RARP, RAPN, and RARC (Figure 1). In total, 47 studies were included for the LC in RARP [9,10,11,12,13,14,15,16,17,18,19,20,21,22,23,24,25,26,27,28,29,30,31,32,33,34,35,36,37,38,39,40,41,42,43,44,45,46,47,48,49,50,51,52,53,54,55], of which 9 were in the last five years [47,48,49,50,51,52,53,54,55] (Table 1). The rest are presented in the Appendix A. For RAPN, 18 studies were included [56,57,58,59,60,61,62,63,64,65,66,67,68,69,70,71,72,73], of which 7 concern the last 5 years [67,68,69,70,71,72,73] (Table 2). The remaining are presented in the Appendix A. Finally, for RARC, 16 studies were included [74,75,76,77,78,79,80,81,82,83,84,85,86,87,88,89], of which 7 were from the last 5 years [83,84,85,86,87,88,89] (Table 3). Appendix A consists of the rest. The overall OT, EBL, and complication rate were evaluated in all three operations. The oncological outcomes of prostatectomy, including positive surgical margins (PSM) and biochemical recurrence (BCR), as well as functional outcomes like incontinence and potency, were evaluated. The LC was also assessed for RAPN regarding WIT and trifecta and for cystectomy regarding lymph node yield. The analysis comprises the studies from 2020, and a comparison with previous studies is made. No studies regarding RARN and RANU were identified.

### 3.1. Robotic-Assisted Radical Prostatectomy

Perioperative outcomes differed among studies, with OT notably decreasing after 100 cases and stabilizing between 200 and 400 cases [48,49,87]. In previous studies, the results followed the same trend, reaching a plateau from 20 to 300 cases [9,11,12,13,14,15,16,17,18,19,22,23,24,25,26,27,29,30,31,34,35,37,38,40,41,43,44]. The same applied to EBL, with the plateau differing between the studies from 100 to 230 cases [47,48,49]. The range was slightly more extensive in previous studies, decreasing from 13 cases to a plateau of 50 to 290 cases [9,11,12,13,14,15,16,18,19,22,23,24,25,26,27,30,34,35,37,38,40,43]. In some studies, the differences with increasing cases were not statistically significant. One recent study observed a plateau at about 200 cases for the length of hospital stay [47]. Earlier studies (Appendix A) likewise suggested that hospital stay tends to stabilize after the initial learning phase with no significant improvement beyond the first few dozen cases. Similarly, previous research indicated improvements in OT and EBL plateau after a specific volume of cases, ranging from roughly 50 up to 300 cases, depending on the study, consistent with the broad ranges noted in our review. In summary, OT, EBL, and hospital stay show steep improvements early in the LC, followed by a plateau once the surgeon gains sufficient experience.

Regarding the oncological results, the plateau for positive margins rate varied highly between the studies, with reports significantly reducing the ratio from 100 cases [49]. In contrast, some studies reported no difference regardless of the case number. Previously, 50 to 300 cases were reported as necessary to achieve these results [9,10,11,13,17,20,23,29,35,38,41,45]. Regarding BCR studies, no association was found with increasing cases. However, in earlier years, a relationship was found between BCR and an increasing number of cases, ranging from 30 to 250 [19,24,25,34]. Recent studies did not evaluate complication rates. In the past, the complication rate was evaluated and was significantly lower after 15–20 cases, with some studies reporting a plateau after 250 cases [13,14,15,23,26,29,31,33,34,35].

Regarding functional outcomes, the plateau for early and 24-month urinary continence reached 200 and 300 cases [51,53]. The same applied in early and 24-month potency rates with improved results after 200 to 300 cases [51,53]. These results follow the previous studies (Appendix A) with the plateau for early and 12-month urinary continence from 100 to 500 cases [29,39]. Twelve-month potency rates were reported from 16 cases in one study [15] to 100 in another [44]. Regarding the trifecta achievement, one study concluded that for three surgeons with previous laparoscopy experience, 20 cases were sufficient [55]. Finally, a study analyzing the LC with specific steps showed that for the entire procedure, 25 cases are needed; for docking, 13 cases; for seminal vesicles dissection, 33 cases; endopelvic fascia dissection, 31 cases; for incising bladder neck, 41 cases, for dissecting prostate and place in bag 38 cases and the most demanding was the urethrovesical anastomosis with 52 cases [50].

### 3.2. Robotic-Assisted Partial Nephrectomy

OT is directly reduced with the increasing number of cases. No plateau is reported, as in most studies, the time continues to lessen the more cases. Most studies did not find statistically significant differences regarding EBL between the case groups. The same conclusion can be assumed for the length of hospital stay with no significant difference, regardless of case number. The same findings were presented in previous years, with one study reporting a plateau in OT at the 16th case [56], one noting a plateau of <120 min OT at the 44th case [65], and that 54 cases were needed for a plateau of <100 mL EBL) [65]. WITs had different LCs among the studies, ranging from 26 to 140 cases [68,69,71]. To accomplish a trifecta between the studies, the number of cases varied between a plateau of 50 and 77 cases [67,68,71]. Regarding safety, it was reported that the complication rate was significantly lower after 20 to 50 cases [68,73]. There was no difference in that range in previous years regarding these three aspects (Appendix A). However, some studies did not show a significant difference or a plateau. Finally, a study found that at least 25 cases are needed to achieve warm ischemia lower than 25 min in complex cases (RENAL score >8) [70].

### 3.3. Robotic-Assisted Radical Cystectomy

OT improved with increasing cases, plateauing from 20 to 75 cases [83,86]. EBL improved in some studies with the increasing number of cases; one reached a plateau of 88 [86]. Regarding the length of hospital stay, the plateau range was between 40 and 198 cases [83,86]. Most of the studies did not show any differences in positive margins. The results agree with those from previous years (Appendix A). Regarding the lymph node yield, no significant difference was observed among studies. Earlier works demonstrated a range from 30 to 50 cases [75,76,77]. The complication rate was reduced with the increasing number of cases, ranging from 40 to 97 needed to reach a plateau [83,84,86,87]. In previous years, the number of cases reached in a range from 16 to 100 cases [78,79,80]. Finally, one study applied the ERUS curriculum and found that a fellow can achieve results equivalent to the expert after 20 cases [88].

### 3.4. Comparing Different Procedure Plateau Case Numbers of the Last Five Years to the Ones of Previous Years

The mean number of cases needed to reach a plateau across all procedures and outcomes was 56.9 in early studies and 67.7 in recent studies (Appendix A). However, no statistically significant difference was observed between these means (t = −1.038, *p* = 0.309). Outcome-specific comparisons (e.g., OT, EBL, length of hospital stay) also revealed no significant differences between early and recent studies.

### 3.5. Risk of Bias Assessment of Included Studies

Common limitations were the observational designs and absence of control groups, leading to a high risk of selection bias (no randomization, though most studies did include consecutive cases) and confounding (Table 4, Appendix A). Few studies adjusted for patient or tumor characteristics, so comparability (confounding) was generally rated as high risk, except in a handful of studies that incorporated concurrent comparison groups or multivariable adjustments (which we judged as moderate risk). Outcome assessment was usually moderate risk: operative metrics (e.g., OT, EBL) were objectively measured, but assessments of complications and functional outcomes were often unblinded and heterogeneous across studies. Selective reporting was a concern in earlier studies—many early LC reports focused on a subset of outcomes (e.g., operative measures) and omitted longer-term functional or oncologic endpoints, yielding a high risk of reporting bias in those cases. New studies tended to report a broader range of outcomes, but since none had pre-registered protocols, we rated most as moderate risk in reporting. In summary, no study was judged to have a low overall risk of bias. The majority were at moderate overall risk, owing primarily to the consistent inclusion of all patients and clearly defined outcomes despite the lack of randomization. A substantial proportion—especially among early single-surgeon case series—were assessed as having a high overall risk of bias due to confounding and limited outcome reporting. These limitations highlight the need to interpret LC results across the included studies carefully.

## 4. Discussion

Advancements in robotic methods in urological surgery have greatly influenced LCs concerning different interventions. A critical aspect examined in our study is whether LCs in robotic surgery have changed over the last five years (2020–2024) compared to previous decades. The overall trend suggests that with technological advancements, improved technical skills and surgical techniques, and structured training programs, LCs appear to have shortened or become more predictable. Towards this direction, robotic surgical simulation-based training has rapidly increased over the past five years, improving LCs and making learning more manageable, even for more complicated urooncological procedures. Additionally, telemedicine-based remote training and support are becoming more and more crucial [90,91,92,93]. An organized, verified, reproducible, and accredited robotic technology deployment is increasingly based on standardized international training pathways. To meet the quickly increasing need for skilled robotic surgeons, a wide range of training programs are still being developed, and teaching approaches are changing. The development of competency criteria for current training methods, the validation of current curricula, and the identification of ways to convert learned abilities in simulation into performance in the operating room and patient outcomes are all opportunities for improvement. Numerous surgical training systems are starting to offer procedure-specific and team training in addition to discrete robotic skills training [93,94,95]. Variability, however, depends on the surgeon’s experience, the volume of operations performed at the facility, and specific difficulties during procedure steps [2]. Interestingly, we demonstrated that despite the evolution of robotic technology and surgical training programs, our analysis indicated that the number of cases required to achieve proficiency has not significantly changed over the past decade. The similarity in plateau case numbers between early and recent studies suggests that the LCs remain stable, and improvements may instead be reflected in finer aspects of surgical performance or complication rates not captured in the current review. These findings support the robustness of current training benchmarks while highlighting the need for additional metrics to evaluate modern surgical learning trajectories.

Recent studies show that after about 200 surgical cases, procedure times in RARP dramatically drop; a following optimization phase occurs at about 400 cases [47]. This suggestion is supported by prior research. In addition, it argues that refinement in surgical procedures with improved haptic feedback and advanced three-dimensional visualization is essential in establishing a threshold level in early performances. Likewise, surgical outcome improvements in PSMs and BCR rates are noted following between 100 and 200 surgical case completion, which aligns with previous studies [45]. Furthermore, functional outcomes in terms of continence also stabilize early (between 200 and 300 cases), with Hashine et al. [53] also documenting a 92% rate at 12-month postsurgical following 200 cases. The advancements are ascribed to developments in robotic systems, like in the case of da Vinci Xi/X models with 3D visualization and improved instrument flexibility, in addition to standardized training incorporating ERUS-developed modules focusing on reducing technical proficiency variability and establishing ideal practices. Finally, surgical technique variations, like the Retzius-sparing RARP (RS-RARP), may demonstrate different reproducibility of different operative steps. A recent multicenter analysis of trainee surgeons reported no significant difference in the proportion of cases achieving a predefined “proficiency” between RS-RARP and standard RARP during the learning phase (43.2% vs. 53.1%, *p* = 0.087) [96]. Trainees performing RS-RARP had shorter OTs and similarly low complication rates, challenging the notion that the RS-RARP is vastly more difficult or time-consuming. It was acknowledged, however, that the RS-RARP cases were undertaken later in the surgeons’ training, suggesting that appropriate timing, case selection, and mentorship can mitigate technical challenges. Consistently, experienced surgeons who transitioned from a standard to a Retzius-sparing technique noted initially longer OTs and more frequent intraoperative urine leak tests with RS-RARP—effects attributed to the technical LC—but without any increase in significant complications [97]. The early postoperative continence advantage of RS-RARP over the standard approach was evident even during the learning period. These findings indicate that although RS-RARP introduces new anatomical nuances and may require technique adjustment, it can be adopted safely without a substantially prolonged LC when surgeons are adequately trained and proctored [98].

Despite notable advancements in RAPN, tumor complexity significantly influences the LC. WIT remains a critical parameter. Research before 2020 suggested that consistency in prolonged WIT plateaus required 100–150 cases, with trifecta outcomes (negative margins, WIT <25 min, no complications) rarely emphasized as endpoints in these studies. The LC for RAPN continues to be a subject of interest as surgeons strive to optimize patient outcomes while navigating the challenges posed by tumor complexity and the need to minimize WIT. Contemporary data, however, propose that WIT improvement may occur as early as 25–50 cases, even for high-complexity tumors (RENAL score >8) [72]. Recent studies [70,73] have shown that the WIT plateau in complex robotic-assisted partial nephrectomy scenarios has decreased to 25–40 cases, significantly reducing from the historical 60–80 cases. By reducing ischemic damage to the kidney during minimally invasive renal surgery [69], advances in robotic instrumentation—including improved vascular sealing and precise suturing techniques—may help explain the improvement in outcomes. These technical and technological improvements have sped the LC and maximized outcomes in this surgical field. Robotic surgery has reached an ample scope and has become an important topic to learn. With the help of robotic surgical technologies, surgeons perform specific complex surgical interventions with much higher accuracy. As a result, these technological advances can help decrease tissue trauma. Trifecta achievement rates now stabilize at 50–77 cases, with Bajalia et al. [67] reporting an 82% success rate post-50 cases compared to 63% in early cohorts. Simulation-based training, including virtual reality (VR) platforms, has reduced early-phase complications by 30–40%, while advancements in robotic instruments (e.g., vascular-selective clamping tools) minimize renal parenchymal damage. Notwithstanding these gains, operating time declines without a distinct plateau, underscoring continuous technical challenges in balancing cancer removal with kidney preservation. Artificial intelligence (AI) for preoperative planning, e.g., tumor segmentation algorithms, further improves accuracy even if cost and access limit general acceptance.

For RARC, the previous ten years have turned a once-prohibitive operation into a reasonable substitute for open surgery. Early series showed complication rates stabilizing around 100 instances, and because of its complexity, intracorporeal urinary diversion (ICUD) was not performed (Appendix A) [79,81]. Still, modern research shows impressive advancements. Compared to 60–80 instances recorded earlier, surgeons educated under regimented courses such as the ERUS program gain ICUD proficiency within 20–40 cases [83]. Additional, high-volume single-center research has demonstrated improved outcomes once the intracorporeal technique is mastered [81,82]. Gaining proficiency in ICUD typically necessitates an increased number of cases; some studies indicate that enhancements in outcomes—such as urinary continence in patients with neobladders or shorter OTs—may be noticed after around 20 to 30 procedures [82]. It is also crucial for surgeons to reconcile the necessity for oncologic precision with the additional OT required for intracorporeal reconstruction during their initial cases [84]. Regarding complication rates, Lombardo et al. [83] demonstrated that Clavien ≥III complications plateau early (40–60 instances), with rates falling from 35% in first cases to 5% post-60 operations. Team-based learning models, emphasizing synchronization between console surgeons and bedside assistants, have further compressed LCs, reducing OT plateaus to 20–30 cases. Lymph node yield and PSM rates remain inconsistently correlated with experience, likely due to the procedure’s inherent complexity and variability in pelvic anatomy. Notably, recent data indicates that after as little as 20 cases, standardized training courses have helped inexperienced surgeons produce results equivalent to specialists [88]. This method differs from past years in which recorded plateaus span more than 50–100 cases [77]. The primary drivers of these developments include technological advancements, formal training programs that replace classical apprenticeship models, and a shared understanding regarding standardized measures (e.g., trifecta, pentafecta). Various challenges do, however, persist. The discrepancy in definitions regarding LC outcomes (e.g., OT vs. trifecta) makes study comparison problematic. In contrast, the unequal distribution of resources leads to imbalances between high-volume versus low-volume centers. For instance, Perera et al. reported a rate of 14.4% PSM in low-volume centers versus 6.1% in high-volume centers [54].

Notwithstanding these developments, this study has certain restrictions. Direct comparison is challenging due to the variation coming from inconsistent definitions of LCs and outcomes reported in various studies. In addition, publication bias can affect outcomes because high-volume centers dominate the literature, which can cause a lack of documentation regarding challenges faced by smaller-volume institutions. To overcome this limitation, researchers have created a sophisticated parameter, the “proficiency score,” namely the coexistence of a comparable operation time to the interquartile range of a mentor, the absence of significant perioperative complications, the absence of perioperative blood transfusions, and the negative surgical margins. The proficiency score has been assessed in RARP and RAPN and predicted better outcome achievement 1-year postoperatively [96,99,100]. One key limitation of this review is the absence of empirical data on RARN and RANU. One explanation could be that these procedures are considered less complex, with no reconstruction steps, and have been adopted earlier than more complex ones [101]; many surgeons may become proficient relatively quickly. Nevertheless, this lack of evidence is significant because the LCs associated with RARN and RANU are not well-defined. This shortage of empirical evidence supports the call for future studies to examine these procedures. Such studies will help determine whether the LC dynamics for RARN and RANU are similar to those of other robotic urological surgeries or if they present unique challenges. Acknowledging this limitation, we emphasize that our conclusions apply only to RARP, RAPN, and RARC, and filling this evidence gap for RARN and RANU is essential to provide a more comprehensive understanding of LCs in all robotic urological oncology procedures [1]. Finally, the moderate to high bias risk in included studies highlights the need for high-quality evidence.

Promising points rely on the rapid advancements in robotic technology, with single-port devices, which can also alter LCs within the timeframe of studies included in this review, hence ongoing monitoring. Future initiatives should prioritize defining agreed-upon LC endpoints, such as trifecta and pentafecta, to enhance comparison between studies. Estimating individual learning routes allows artificial intelligence and machine learning approaches to be tailored to specific needs and maximize competency attainment. Moreover, economic assessments contrasting organized training courses with conventional apprenticeship schemes could offer insightful analysis of the cost-effectiveness of several learning strategies. Although robotic-assisted urological surgery is still developing, our results highlight the consistency in LCs irrespective of the type of operation and surgeon background. Meanwhile, the past five years (2020–2024) have seen a notable reduction in learning times, driven by advancements in robotic technology. Standardizing LC evaluation measures will be vital if we maximize training paths and guarantee the best patient outcomes.

## 5. Conclusions

The evolution of robotic urological surgery over the past five years (2020–2024) has demonstrated significant advancements in LCs. While LCs remain procedure-specific and inconsistently defined, recent data indicate notable improvements in proficiency acquisition compared to earlier decades. The last decade has redefined LCs in robotic urological surgery, with different robotic urooncological procedures demonstrating 30–50% reductions in case requirements. With reduced LCs, the last five years have confirmed robotic surgery as a pillar of urological oncology.

## Figures and Tables

**Figure 1 cancers-17-01334-f001:**
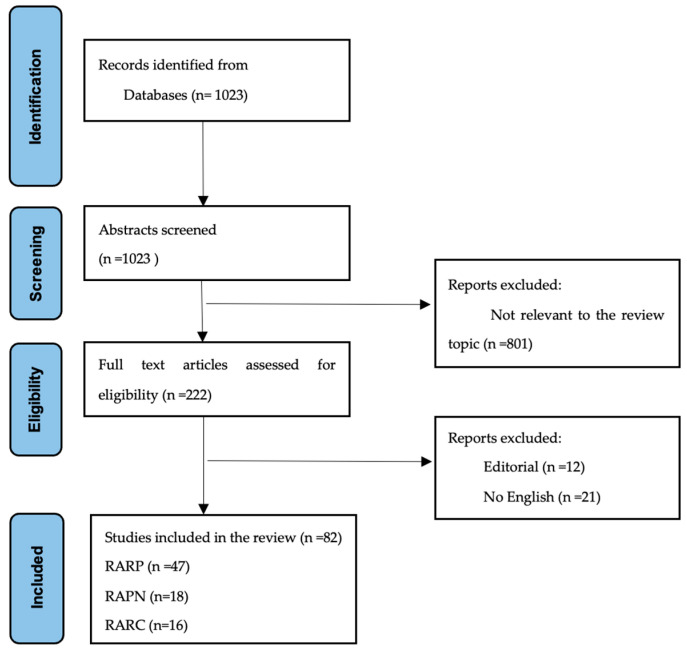
PRISMA search strategy for selection of related and relevant research.

**Table 1 cancers-17-01334-t001:** Summary of studies assessing learning curves in robotically assisted radical prostatectomy over the previous five years.

Author (Year)	Number (Patients)	Number (Surgeons)	Prior Experience	Main Peri-Operative Outcomes			Main Oncological Outcomes		Safety Outcomes	Main Functional Outcomes	
				Operative Time	Estimated Blood Loss	Length of Stay	PSM Rate	BCR Rate	Outcomes	Continence Rate	Potency Rate
Chen (2020) [47]	500	1	Open radical prostatectomy	First plateau 200 cases; second decrease after 400 cases	First plateau 200 cases; second decrease after 400 cases	Plateau after 200 cases.	No statistically significant differences between groups	N/A	N/A	N/A	N/A
Song (2020) [48]	480	1	Novice in RARP	Plateau reached at 200th case (35 month)	Plateau reached at 230th case (37 months)	N/A	N/A	N/A	N/A	N/A	N/A
Baunacke (2021) [49]	703	3	Open radical prostatectomy	<100 cases: 233.7 min >100 cases: 184.1 min	<100 cases: 888.4 mL >100 cases: 604.2 mL	N/A	<100 cases: 21% >100 cases: 15%	N/A	N/A	N/A	N/A
Ambinder (2022) [50]	120	1	Fellowship program	N/A	N/A	N/A	N/A	N/A	N/A	N/A	N/A
Bock (2022) [51]	2672	25	N/A	N/A	N/A	N/A	<75 cases 21% >300 cases 24%	<75 cases 11% >300 cases 13%	N/A	3-month: <75 cases 54% >300 cases 44% 24-month: <75 cases 21% >300 cases 16%	3-month: <75 cases 20% >300 cases 36% 24-month: <75 cases 38% >300 cases 53%
Gandi (2022) [52]	761	3	N/A	N/A	N/A	N/A	Surgeon A (mentor) 153 cases Surgeon B: 12 cases Surgeon C 31 cases	N/A	N/A	N/A	N/A
Hashine (2023) [53]	319	7	N/A	N/A	N/A	N/A	Not significant difference between groups (0–100, 100–200, >100)	Not significant difference between groups (0–100, 100–200, >100)	N/A	Better results after 200 cases	Better results after 200 cases
Perera (2023) [54]	3969- 556 operated by surgeons who performed <50 RARP-general cohort surgeons	53	N/A	Mean operative time: 266 min for general cohort surgeons 240 min for surgeons in high volume centers	Mean estimate blood loss: 361 mL for general cohort surgeons 302 mL for surgeons in high volume centers	N/A	14.4% for general cohort surgeons 6.1% for surgeons in high volume centers	N/A	N/A	N/A	N/A
Carlos (2024) [55]	146	3	Laparoscopy	N/A	N/A	N/A	N/A	N/A	N/A	N/A	N/A

PSM positive margins. BCR biochemical recurrence. N/A not available.

**Table 2 cancers-17-01334-t002:** Summary of studies assessing learning curves in robotically assisted partial nephrectomy over the previous five years.

Author (Year)	Number (Patients)	Number (Surgeons)	Prior Experience	Main Peri-Operative Outcomes			Trifecta	Safety Outcomes	Warm Ischemia Time
				Operative Time	Estimated Blood Loss	Length of Stay			
Bajalia (2020) [67]	406	1	N/A	1–50 cases: 223 min 51–100 cases: 204 min 101–150 cases: 202 min 151–200 cases: 201 min 201–250 Cases: 196 min 251–300 cases: 188 min 301–350 cases: 194 min 351–400 cases: 197 min >400 cases: 186 min	N/A	N/A	1–50 cases: 63% 51–100 cases: 82% 101–150 cases: 66% 151–200 cases: 67% 201–250 cases: 54% 251–300 cases: 71% 301–350 cases: 84% 351–400 cases: 74% >400 cases: 72% Plateau 77 cases	High grade complication: 1–50 cases: 8% 51–100 cases: 4% 101–150 cases: 9% 151–200 cases: 10% 201–250 cases: 8% 251–300 cases:2% 301–350 cases: 6% 351–400 cases: 6% >400 cases: 6%	1–50 cases: 16.5 min 51–100 cases: 13.4 min 101–150 cases: 16.7 min 151–200 cases: 17.1 min 201–250 cases: 19.6 min 251–300 cases: 16.7 min 301–350 cases: 18.2 min 351–400 cases: 16.6 min >400 cases: 17.7 min
Castilho (2020) [68]	101	1	N/A	1–50 cases: 114 min 51–100 cases: 120 min	1–50 cases: 295 mL 51–100 cases: 375 ml	N/A	1–50 cases: 58% 51–100 cases: 87.8%	Complication rate:1–50 cases:18% 51–100 cases: 8%	1–50 cases: 17.3 min 51–100 cases: 11.7 min
Motoyama (2020) [69]	65	1	RARP	0–13 cases: 140–150 min 14–26 cases: 130–140 min 27–39 cases: 110–120 min 40–52 cases: 120–130 min 53–65 cases: 110–120 min	Median estimate blood loss: 50 mL	Median length of stay: 9 days	N/A	N/A	0–13 cases: 19 min 14–26 cases: 16 min 27–39 cases: 17 min 40–52 cases: 15 min 53–65 cases: 13 min
Fiorello (2021) [70]	172 experts (44–55-45_ training surgeons	4 1 expert 3 training surgeons	N/A	N/A	N/A	N/A	N/A	N/A	N/A
Zeuschner (2021) [71]	500	N/A	N/A	1–143 cases: 172 min 144–500 cases: 152 min	1–143 cases: 220 mL 144–500 cases: 200 mL	1–143 cases: 7 days 144–500 cases: 6 days	1–143 cases: 53.8% 144–500 cases: 60.8%	Complication rate: 1–143 cases: 30.1% 144–500 cases: 22.1%	1–143 cases: 18 min 144–500 cases: 13 min
Al-Nader (2023) [72]	127	2	N/A	1–18 case: 242 min 19–38 case: 208 min >39 cases: 109 min	N/A	N/A	N/A	N/A	N/A
Zhang (2023) [73]	50	1	N/A	1–24 cases: 133.5 min 25–50 cases: 115.31 min	1–24 cases: 117.92 mL 25–50 cases: 120.38 mL	1–24 cases: 5.33 days 25–50 cases: 4.3 days	1–24 cases: 22/24 (91.7%) 25–50 cases: 21/26 (81.8%)	Complication rate: 1–24 cases: CD < 2: 95.8% CD ≥ 2: 93.1% 25–50 cases: CD < 2: 4.2% CD ≥ 2: 6.9%	N/A

N/A not available.

**Table 3 cancers-17-01334-t003:** Summary of studies assessing learning curves in robotically assisted radical cystectomy over the previous five years.

Author (Year)	Number (Patients)	Number (Surgeons)	Prior Experience	Main Peri-Operative Outcomes			PSM Rate	Safety Outcomes	Lymph Node Yield
				Operative Time	Estimated Blood Loss	Length of Stay			
Lombardo (2021) [83]	100	1	N/A	Plateau reached at 20th case; 0–10 cases: 640 min	<40 cases: drop of Hb >2 g/dL over 30% >40 cases: drop of Hb >2 g/dl under 30%	Plateau reached at 40th case	PSM do not change significantly along the LC	Complication rate: 0–20 cases: 35% 20–40 cases: 20% 40–60 cases: 15% 60–80 cases: 5% 80–100 cases: 0%	Number of LNs did not change along the LC
Tuderti (2021) [84]	137	1	N/A	0–45 cases: 337.6 min 46–90 cases: 339.1 min 91–137 cases: 282.5 min	Number of intraoperative transfusions: 0–45 cases: 2 46–90 cases: 2 91–137 cases: 3	0–45 cases: 17.5 46–90 cases: 12.3 91–137 cases: 11.9	0–45 cases: 2 46–90 cases: 1 91–137 cases: 1	0–45 cases: 68.9% 46–90 cases: 28.2% 91–137 cases: 17.4%	0–45 cases: 28.9 46–90 cases: 29.2 91–137 cases: 29.3
Lopez Molina (2022) [85]	62	3	N/A	0–20 cases: 398.5 min 21–40 cases: 315.3 min 41–62 cases: 337.4 min	Number of intraoperative transfusions: 0–20 cases: 2 21–40 cases: 0 41–62 cases: 1	0–20 cases: 10 days 21–40 cases: 9 days 41–62 cases: 11.5 days	0–20 cases: 0 21–40 cases: 1 41–62 cases: 1	Complication rate: 0–20 cases: 75% (15) 21–40 cases: 75% (15) 41–62 cases: 81.8% (18)	0–20 cases: 20 21–40 cases: 17 41–62 cases: 15.5
Wijburg (2022) [86]	2186	N/A	N/A	Plateau reached after 75 cases (321 min)	Plateau reached after 88 cases (292 mL)	Plateau reached after 198 cases (9.5 days)	N/A	Plateau reached after 97 cases (48%)	N/A
Achermann (2023) [87]	53	1	N/A	0–14 cases: 415 min 15–27 cases: 390 min 28–40 cases: 445 min 41–53 cases: 361 min	0–14 cases: 400 mL 15–27 cases: 300 mL 28–40 cases: 300 mL 41–53 cases: 200 mL	0–14 cases: 16 days 15–27 cases: 16 days 28–40 cases: 22 days 41–53 cases: 16 days	N/A	Complication rate overall: 0–14 cases: 79% 15–27 cases: 69% 28–40 cases: 85% 41–53 cases: 38%	0–14 cases: 19 15–27 cases: 29 28–40 cases: 19 41–53 cases: 23
Diamand(2023) [88]	N/A	1	Erus curriculum	N/A	N/A	N/A	N/A	N/A	
Tuderti (2024) [89]	200	2	N/A	0–66 cases: 342 min 67–133 cases: 316 min 134–200 cases: 319 min	N/A	0–66 cases: 14.9 days 67–133 cases: 11.1 days 134–200 cases: 6.8 days	N/A	N/A	N/A

PSM positive margin. N/A not available. Hb hemoglobin. LC learning curve. LNs lymph nodes.

**Table 4 cancers-17-01334-t004:** Risk of bias assessment of included studies.

Study (Author, Year)	Selection Bias	Comparability (Confounding)	Outcome Assessment	Reporting Bias	Overall RoB
RARP—Recent (Last 5 Years)					
Chen (2020) [47]	Moderate	High	Moderate	Moderate	Moderate
Song (2020) [48]	Moderate	High	Moderate	Moderate	Moderate
Baunacke (2021) [49]	Moderate	Moderate	Moderate	Moderate	Moderate
Ambinder (2022) [50]	Moderate	High	Low	Moderate	Moderate
Bock (2022) [51]	Low	High	Moderate	Moderate	Moderate
Gandi (2022) [52]	Moderate	Moderate	Low	Moderate	Moderate
Hashine (2023) [53]	Moderate	High	Moderate	Moderate	Moderate
Perera (2022) [54]	Moderate	High	Moderate	Moderate	High
Carlos (2024) [55]	Moderate	High	Moderate	Moderate	High
RAPN—Recent (Last 5 Years)					
Bajalia (2020) [67]	Moderate	High	Moderate	Moderate	High
Castilho (2020) [68]	Moderate	High	Moderate	Moderate	High
Motoyama (2020) [69]	Moderate	High	Moderate	Moderate	High
Fiorello (2021) [70]	Moderate	High	Moderate	Moderate	High
Zeuschner (2021) [71]	Moderate	Moderate	Moderate	Moderate	Moderate
Al-Nader (2023) [72]	Moderate	High	Moderate	Moderate	High
Zhang (2023) [73]	Moderate	High	Moderate	Moderate	High
RARC—Recent (Last 5 Years)					
Porreca (2020) [82]	Moderate	High	Moderate	Moderate	High
Tuderti (2020) [84]	Moderate	High	Moderate	Moderate	High
Lombardo (2021) [83]	Moderate	High	Moderate	Moderate	Moderate
López-Molina (2021) [85]	Moderate	High	Moderate	Moderate	High
Wijburg (2022) [86]	Moderate	Moderate	Moderate	Moderate	Moderate
Achermann (2023) [87]	Moderate	High	Moderate	Moderate	High
Diamand (2023) [88]	Moderate	Moderate	Moderate	Moderate	Moderate
Tuderti (2024) [89]	Moderate	High	Moderate	Moderate	High

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
