# Peer review of "Learning Curves in Robotic Urological Oncological Surgery: Has Anything Changed During the Last Five Years?"

_cancers, 2025, doi:10.3390/cancers17081334_

Round 1
Reviewer 1 Report
Comments and Suggestions for Authors
I read with great interest this manuscript on the learning curves in robotic urological oncological surgery.
This work reviews the current literature for available studies regarding the learning curves for robotic surgery in surgical oncology in urology.
Below are my suggestions to improve the quality of this review:
First is this a systematic o narrative review? if systematic where is the PRISMA figure?
Second, Table 1 on the main outcomes of the studies , divided by procedure, would be appreciated.
Third, why a paragraph on radical nephrectomy was not included.
Lastly about radical prostatectomy, some studies have hypothesized a different learning curve between Retzius sparing vs standard RARP performed by trainee surgeons. doi: 10.5173/ceju.2023.260. authors may highlight this topic in the discussion section.
Looking forward by receiving the second version of the manuscript.
With Regards
Comments on the Quality of English Language
no comments on this section
Author Response
I read with great interest this manuscript on the learning curves in robotic urological oncological surgery.
This work reviews the current literature for available studies regarding the learning curves for robotic surgery in surgical oncology in urology.
Below are my suggestions to improve the quality of this review:
Comment 1: First is this a systematic o narrative review? if systematic where is the PRISMA figure?
Response (Lines: 70, 86-89, 113-141): We thank the reviewer for raising this critical point. We apologize for the confusion regarding the review methodology. In the revised manuscript, we have clarified that our study is a systematic literature search and narrative review of the learning curve literature rather than a traditional systematic review with meta-analysis. We used defined search terms in MEDLINE and followed PRISMA guidelines for the search strategy and selection process (and have included a PRISMA-style flow diagram in Figure 1). Still, we did not perform a quantitative meta-analysis. Because we aimed to summarize heterogeneous studies qualitatively, many elements of PRISMA (e.g., meta-analytic pooling of data) and a formal risk-of-bias scoring were not fully applicable. We have now explicitly stated in the Methods that this is a “systematic search and narrative literature review” to distinguish it from a full systematic review.
Comment 2: Second, Table 1 on the main outcomes of the studies , divided by procedure, would be appreciated.
Response: We thank the reviewer for this comment. Also, according to the editors’ suggestion, we have added three tables (Table 1,2,3), one for each procedure (radical prostatectomy, partial nephrectomy, and radical cystectomy). The studies included in these tables were conducted during the last five years. Additionally, the manuscript contains three supplementary tables from older studies, one for each procedure. (Supplementary tables 1,2,3)
Comment 3: Third, why a paragraph on radical nephrectomy was not included.
Response (Lines: 110-11, 366-377): We thank the reviewer for this comment.
Robotic Radical Nephrectomy (RARN): As noted in the Results, our systematic search did not identify any specific studies on the learning curve for robot-assisted radical nephrectomy​, but also on robot-assisted radical nephroureterectomy. We have clarified this in the Discussion and added context as to why this might be the case. Specifically, we note that radical nephrectomy is generally a less complex procedure (compared to partial nephrectomy or cystectomy) and has been one of the earlier adopted robotic procedures; many surgeons may become proficient relatively quickly, which could explain the lack of dedicated RARN learning curve studies. We added a reference to support this assumption. (Xiong S, Jiang M, Jiang Y, Hu B, Chen R, Yao Z, Deng W, Wan X, Liu X, Chen L, Fu B. Partial Nephrectomy Versus Radical Nephrectomy for Endophytic Renal Tumors: Comparison of Operative, Functional, and Oncological Outcomes by Propensity Score Matching Analysis. Front Oncol. 2022 Jul 26;12:916018. doi: 10.3389/fonc.2022.916018. PMID: 35957884; PMCID: PMC9360524)
Comment 4: Lastly about radical prostatectomy, some studies have hypothesized a different learning curve between Retzius sparing vs standard RARP performed by trainee surgeons. doi: 10.5173/ceju.2023.260. authors may highlight this topic in the discussion section.
Response (Lines: 281-298): We thank the reviewer for this critical comment.
We added more information about this topic in the robot-assisted radical prostatectomy discussion paragraph.
Looking forward by receiving the second version of the manuscript.
With Regards
Reviewer 2 Report
Comments and Suggestions for Authors
Authors have to be commended for evaluating the learning curve in urological robotic surgery. The learning curve is a crucial issue, and reviewing the last five years of literature is highly valuable. The manuscript follows a structured methodology using PRISMA guidelines, covering multiple procedures (RARP, RAPN, RARC) and key learning parameters such as operative time, blood loss, complications, functional outcomes.
Comments:
- introduction should be improved and the aim defined more precisely; consider to underline the lack of general consensus on learning curve definition
- highlight as a significant limitation the lack of studies on RARN and RANU
- to improve readibility, provide a table summarizing plateau case numbers across different procedures
- provide a risk of bias assessment. Did the authors weigh the importance of high-volume vs. low-volume center studies? This is a relevant issue being the plateau not consistent across different centers and for different studies.
- consider to discuss studies about proficiency score as a surrogate for surgical quality and plateau (e.g. PMID 39176300; PMID 38673499; PMID 37064261)
Author Response
Authors have to be commended for evaluating the learning curve in urological robotic surgery. The learning curve is a crucial issue, and reviewing the last five years of literature is highly valuable. The manuscript follows a structured methodology using PRISMA guidelines, covering multiple procedures (RARP, RAPN, RARC) and key learning parameters such as operative time, blood loss, complications, functional outcomes.
We thank the reviewer for the nice comments and tried to assess the requests separately.
Comments:
Comment 1: Introduction should be improved and the aim defined more precisely; consider to underline the lack of general consensus on learning curve definition
Response (Lines: 47-52, 60-67): We thank the reviewer for this essential comment. We enriched the introduction in a way to highlight the lack of consensus not only on the learning curve definition but also on the different parameters defining a learning curve.
Comment 2: Highlight as a significant limitation the lack of studies on RARN and RANU
Response (Lines 110-111, 366-377): We thank the reviewer for this comment, which was also a request of another reviewer. We have, therefore, added a limitation with possible explanations and future research needs in the Discussion.
Comment 3: To improve readability, provide a table summarizing plateau case numbers across different procedures
Response (Lines: 90-97, 210-217, 260-268): We thank the reviewer for this critical comment. We have added a supplementary table 4, containing plateau case numbers for robotic urologic procedures across included studies along with the comparison of the last 5 years with the previous evidence, and we did not find any significant difference. We added information in the Methods, Results, and Discussion to support these data.
Comment 4: Provide a risk of bias assessment. Did the authors weigh the importance of high-volume vs. low-volume center studies? This is a relevant issue being the plateau not consistent across different centers and for different studies.
Response (Lines: 219-239, 377-378): Again, we thank the reviewer for this comment. Regarding the risk of bias assessment, we agree that transparency about study quality is essential. Instead of a formal risk-of-bias tool (which is challenging given that most included articles are retrospective case series without a uniform quality appraisal tool), we have added a summary Table 4 in the revised manuscript that outlines key aspects of each study, including study design, the clinical setting (e.g., single vs. multi-center, surgeon experience level), and significant limitations or potential biases. This serves as a risk-of-bias overview for the included studies. For example, nearly all included studies are retrospective case series or cohort studies, often single-surgeon or single-center experiences, which inherently carry risks of selection bias and limited generalisability. We have highlighted such limitations in the new table and the Discussion. We believe this addition addresses the reviewer’s concern by providing readers with a concise “at-a-glance” assessment of the evidence base’s quality. (Please see Supplementary Table 5 below for the summary of included studies’ quality and limitations.) We hope these clarifications and additions adequately explain the nature of our review and our rationale for not applying meta-analytic tools or formal bias scoring in this context. We remain open to further guidance on this point if needed.
Comment 5: Consider to discuss studies about proficiency score as a surrogate for surgical quality and plateau (e.g. PMID 39176300; PMID 38673499; PMID 37064261)
Response (Lines: 360-366): We thank the reviewer for this final comment. We have added a short paragraph to the Discussion about the significance of the proficiency score in predicting optimal outcomes after robotically-assisted radical prostatectomy and partial nephrectomy.
Reviewer 3 Report
Comments and Suggestions for Authors
The authors present a systematic review regarding the learning curves for robotic surgery in urological oncology. Also, they compare the learning curves indicated by the studies of the last 5 years to those indicated by previously published results.
The use of English is appropriate. However, some sentences require refinement.
Results
Line 132-135 : The phrase “with one study reporting a plateau at the 16th case…” should specify which surgical outcome this plateau refers to.
Line 147-148 : The authors should also present the results from the previous studies regarding the learning curves in association with OT, EBL and length of hospital stay.
Regarding RARC, the authors should discuss the type of urinary diversion used in the studies included in the review and the potential impact of the type of diversion on the learning curves.
Overall , this is a well-written systematic review that shares critical insights into the improvements in learning curves for robotic surgery in urological oncology over the recent years.
Comments on the Quality of English LanguageThe use of English is appropriate. However, some sentences require refinement.
Author Response
The authors present a systematic review regarding the learning curves for robotic surgery in urological oncology. Also, they compare the learning curves indicated by the studies of the last 5 years to those indicated by previously published results.
Comment 1 : The use of English is appropriate. However, some sentences require refinement.
Response (Corrections in yellow text highlight color): We thank the reviewer for this comment. We have made language polishing by using the Grammarly language editing software. We believe that the language has been significantly improved after this editing.
Results
Comment 2: The phrase “with one study reporting a plateau at the 16th case…” should specify which surgical outcome this plateau refers to.
Response (Lines: 186-188): We thank the reviewer for this comment and made the requested amendment. In this revised sentence, we have specified the surgical outcome for the plateau at the 16th case as operative time (OT). This clarification makes it clear that the study referenced in [56] found its learning curve plateau (i.e., the point of proficiency) at 16 cases concerning operative time.
Comment 3: The authors should also present the results from the previous studies regarding the learning curves in association with OT, EBL and length of hospital stay.
Response (Lines: 151-158): We thank the reviewer for this comment. We added this information in the robotically-assisted radical prostatectomy paragraph in the Results. We also have a supplementary Table 1 that illustrates results from older studies for a more thorough search by the readers. In this expanded text, we have incorporated findings from previous studies regarding operative time, EBL, and hospital stay: earlier literature reported that most of the gains in reducing OT and EBL occur within the first dozens of cases, after which these metrics plateau (with reported plateau points ranging from tens to a few hundred cases). We also note that length of stay shows a similar pattern of early improvement, often levelling off relatively soon (no major decrease in hospital stay after the surgeon’s first 20–50 cases in some reports). This provides context from prior studies to complement the single-study finding of a 200-case plateau for hospital stay, thereby addressing the reviewer’s request to include previous studies’ observations for these perioperative outcomes.
Comment 4: Regarding RARC, the authors should discuss the type of urinary diversion used in the studies included in the review and the potential impact of the type of diversion on the learning curves.
Response (Lines: 329, 332-338): Again, we thank the reader for this valuable comment. We have added a paragraph in the results discussing how different types of urinary diversion in robotic radical cystectomy (RARC) can influence the learning curve. By including this discussion, we acknowledge that not all “RARC” learning curves are equivalent—those incorporating intracorporeal urinary diversion are inherently steeper and require more cases to achieve proficiency.
We hope this addition meets the reviewer’s request and adds valuable nuance for readers interested in RARC.
Overall , this is a well-written systematic review that shares critical insights into the improvements in learning curves for robotic surgery in urological oncology over the recent years.
Comment 5: The use of English is appropriate. However, some sentences require refinement.
Response (Corrections in yellow text highlight color): We thank the reviewer for this comment. We have done language polishing by using the Grammarly language editing software. We believe that the language has been significantly improved after this editing.
Round 2
Reviewer 2 Report
Comments and Suggestions for Authors
Authors provided a point-to-point reply, the manuscript was modified accordingly. No improvements needed. My recognition for the good job done.